# Computational and Experimental Tools to Monitor the Changes in Translation Efficiency of Plant mRNA on a Genome-Wide Scale: Advantages, Limitations, and Solutions

**DOI:** 10.3390/ijms20010033

**Published:** 2018-12-21

**Authors:** Irina V. Goldenkova-Pavlova, Olga S. Pavlenko, Orkhan N. Mustafaev, Igor V. Deyneko, Ksenya V. Kabardaeva, Alexander A. Tyurin

**Affiliations:** 1Group of Functional Genomics, Institute of Plant Physiology, Russian Academy of Sciences, Botanicheskaya str. 35, Moscow 127276, Russia; helliga.p@gmail.com (O.S.P.); igor.deyneko@inbox.ru (I.V.D.); kabardaewa@yandex.ru (K.V.K.); alexjofar@gmail.com (A.A.T.); 2Department of Biophysics and Molecular Biology, Baku State University, Zahid Khalilov Str. 23, Baku AZ 1148, Azerbaijan; orkhan@bioset.org

**Keywords:** regulation and efficiency of translation, genome-wide scale, experimental approaches, computational algorithms, features of plant mRNAs

## Abstract

The control of translation in the course of gene expression regulation plays a crucial role in plants’ cellular events and, particularly, in responses to environmental factors. The paradox of the great variance between levels of mRNAs and their protein products in eukaryotic cells, including plants, requires thorough investigation of the regulatory mechanisms of translation. A wide and amazingly complex network of mechanisms decoding the plant genome into proteome challenges researchers to design new methods for genome-wide analysis of translational control, develop computational algorithms detecting regulatory mRNA contexts, and to establish rules underlying differential translation. The aims of this review are to (i) describe the experimental approaches for investigation of differential translation in plants on a genome-wide scale; (ii) summarize the current data on computational algorithms for detection of specific structure–function features and key determinants in plant mRNAs and their correlation with translation efficiency; (iii) highlight the methods for experimental verification of existed and theoretically predicted features within plant mRNAs important for their differential translation; and finally (iv) to discuss the perspectives of discovering the specific structural features of plant mRNA that mediate differential translation control by the combination of computational and experimental approaches.

## 1. Introduction

The genomic information in plants, similar to other eukaryotes, is implemented via a successive series of biological processes, including transcription and translation as the key events. The current experimental omics tools for genomic monitoring of plant gene expression allow tracking the flow of genetic information from genome to proteome and to metabolome. New experimental approaches, for example, RNA-Seq and DNA microarrays, have given insight into many key mechanisms involved in transcription regulation in plants: the first stage of gene expression and the easiest to study in terms of experimental methodology. The studies of transcriptomes, i.e., the qualitative and quantitative estimation of expression of the entire gene pool on a genome-wide scale, have given convincing evidence of dynamic changes in the transcriptomes of various plant species in both growth and development processes and the impact of environmental factors. Comparative omics studies in plants clearly demonstrate a very modest correlation between the levels of transcription (abundance of individual mRNAs) and translation (the levels of the corresponding proteins in the proteome). Of note, the observed fluctuations in the levels of a transcript do not always lead to changes in the levels of the corresponding protein [1]. This suggests an intricate nature of the mechanisms providing the decoding of a genome, which involve not only differential transcription, but also differential translation.

Translation is a complex biological process with numerous players, including mRNAs, tRNAs, ribosomes, and manifold protein factors. Undoubtedly, each is important for efficient translation. The mRNAs themselves comprise different regions, namely, the 5’ untranslated region (5’UTR) and coding region (CDS) and 3’ untranslated region (3’UTR), which modulate translation at a number of “checkpoints”: translation initiation, elongation, and termination. In the current view, numerous regulatory elements may be concealed in the nucleotide contexts of these mRNA regions and each of them individually or in combination can determine the development of any transcript in translational process [2].

The paradox of misfit between the levels of mRNAs and their protein products observable in different plant species at all stages of their growth and development as well as upon the impact of various environmental factors focuses the attention of researchers on two key problems, namely (i) detection of the specific sets of differentially-translated transcripts, i.e., the sets of transcripts that are efficiently translated under certain conditions, and the sets of transcripts with repressed or unchanged translation under the same conditions and (ii) clarification of the particular regions or specific structural features of the mRNA nucleotide composition that mediate this differential translational control.

This review focuses on the experimental methods for genome-wide analysis of translational control, computational algorithms to search and analyze various regulatory contexts within mRNAs, and approaches for subsequent experimental verification of their correlation with mRNA translation in plants. Currently, we cannot refer to deficiency in publications comprehensively reporting the basic protocols of various methods for genome-wide analyses of translational control in general, including the methods applicable to plant objects. However, reviews that consider and discuss the three key components of the general strategy for identification of regulatory contexts in mRNA that may play a key role in differential translation are still absent in the scientific literature. Our goals here are (i) to consider the experimental approaches aiming to clarify differential translation on a plant genome-wide scale; (ii) to summarize the current data on the computational algorithms used for detection of the specific structural and functional features of key determinants within plant mRNAs and their interrelation with the translation efficiency; (iii) to highlight the methods for experimental verification of existed data and theoretical predictions of the intrinsic features of plant mRNAs important for their differential translation; and (iv) to discuss the ways of decoding the specific structural features of plant mRNA that mediate differential translational control by combining computational and experimental approaches. In general, this review discusses the main and critical steps for each method in this general strategy, areas of their application, and the main results obtained using plant objects and their contribution to our knowledge about the fine mechanisms of translation in plants.

## 2. Experimental Approaches to Determine Differentially-Translated mRNAs in Plants

Initially, proteomics methods were used to identify the correlation between the observed fluctuations in the expression of a transcript and the actual level of peptides in plants [3]. However, the proteomics approaches have certain limitations in the case of a spatiotemporal study of a large pool of translated mRNAs and are mainly applied for assessing translation of the known peptides and proteins. Moreover, the methods of proteomics are laborious and expensive, while preparation of the specimens, quantification of proteome, and subsequent peptide sequencing require specialized technical experience [4]. Advances in high-throughput technologies, such as microarrays and deep sequencing, have made it possible to develop the new experimental approaches to studying mRNA translation efficiency on a global scale. Three basic experimental approaches are currently used for these purposes: (a) polysome profiling; (b) translating ribosome affinity purification (TRAP); and (c) ribosome profiling or Ribo-Seq. These approaches are based on (i) the production of the mRNA pool with the ribosomes arrested on them; (ii) separation of actively-translated mRNAs (polysomal mRNAs and mRNAs bound to several ribosomes), moderately translated mRNAs (monosomal mRNAs and mRNAs bound to one ribosome), and untranslated mRNAs (steady-state mRNAs that are not bound to ribosomes); and (iii) subsequent quantitative assessment of an individual transcript or an mRNA population represented in polysomal complexes relative to the total amount of the transcript in the assayed plant specimens. Note that polysomes are several ribosomes performing translation from one mRNA and this process is regulated for individual mRNAs.

### 2.1. Profiling Polysomes

The translational status of the mRNA pool on a genome-wide scale can be estimated using polysome profiling. The basic protocols for the polysome profiling in plants are described in several publications [5]. Simple protocols have been additionally designed and verified for individual plant species, including *Arabidopsis thaliana*, *Nicotiana benthamiana*, *Solanum lycopersicum*, and *Oryza sativa*, as well as for individual plant tissues [5]. This method is based on the separation of the polysomal mRNAs, monosomal mRNAs, and steady-state mRNAs using sucrose density gradient centrifugation, referred to as polysome fractionation assays (Figure 1). Then, the transcripts (mRNAs) associated with each mRNA pool are analyzed by hybridization on microarrays or undergo RNA sequencing. Assembly, mapping, and in silico analysis of the sequencing data for different pools (polysomes, monosomes, and steady-state mRNAs) provide the researcher with the initial lists of the transcripts with different translation activities [6,7].

According to the experimental data, the results of polysome profiling can be used for a quantitative estimation of mRNA translation efficiency both at different plant growth and developmental stages and under the effect of adverse environmental factors [6,8] or for assessment of quantitative changes in the translational status of individual mRNAs [9]. As a rule, the polysome score (PS) or polysome ratio (PR) are used for this purpose; they are computed as the relative abundances of RNAs in polysomes versus RNAs in nonpolysomes or versus total mRNA. Where total mRNA is the total mRNA level in polysomal and nonpolysomal fractions, respectively [6,8,9].

The polysome profiling appeared rather efficient in the studies on differential translation regulation of specific plant mRNAs under the influence of several abiotic environmental factors [2,6,10]. For example, it has been convincingly demonstrated that the main part of the transcripts under stress displays different degrees of translation repression; moreover, a specific set of transcripts that avoids such repression and retain their transcriptional activity was detected. Below are several examples that in our view, illustrate the abilities of this method in clarifying the mechanisms of translation control in plants. In particular, it is shown that the shares of individual mRNA species in *A. thaliana* polysomal fractions under controlled growth upon a moderate dehydration stress vary from 5% to 95% and that this stress causes a decrease in the ribosome load for over 60% of all mRNAs [2]. The results of genome-wide assay of the relative amounts of individual mRNAs in polysomal versus nonpolysomal fractions under heat shock in the *A. thaliana* cell culture gave the set of genes with different translational responses, i.e., the genes that either considerably increased or considerably decreased the amounts of their mRNAs in polysomal fractions [10]. These results formed the background for further identification of the new cis-regulatory elements in 5’UTRs that influenced differential translation in response to heat shock in *A. thaliana* [8].

In another study, polysome profiling was used for a global assessment of the translation efficiency of mRNA pools during the growth and development of *A. thaliana* leaves. It was demonstrated that the degree of association of each mRNA with the polysomal fraction was different and considerably (from a strong repression to activation at a constant level) changed throughout these processes. Analysis of the functional categories of the mRNAs associated with polysomal fraction showed that the translation control, being of physiological significance during plant growth and development, was especially pronounced in the mRNAs associated with signaling and protein synthesis. In general, these results emphasize the importance of the dynamic changes in mRNA translation during plant growth and development and suggest that mRNA translation may be controlled via complex mechanisms underlying the response to each factor [6].

Although polysome profiling has been successfully used for a global study of plant mRNA translation efficiency, this method still has some limitations [11]. One of these, it cannot precisely determine the ribosome density, i.e., the number of ribosomes per mRNA, because the mRNA–ribosome complexes from the same differential centrifugation fractions may contain a different number of ribosomes. Moreover, polysome profiling fails to determine the actual ribosome distribution along the transcript, i.e., it is impossible to determine a mRNA region (5’UTR, CDS, or 3’UTR) in which reside the arrested ribosomes. This is very important since it allows for assessing of the translation stage (initiation, elongation, or termination) associated with differential translation of an individual transcript. As a consequence, this makes it not possible to specifically search for the regulatory determinants in particular mRNA regions important for an efficient translation.

Nonetheless, these limitations of the polysome profiling technique do not diminish its tremendous potential for the study of the fine mechanisms of translation in plants on a global scale. This method not only makes it possible to determine the correlations between the observed translational and transcriptional fluctuations under normal conditions and under stress factors, but also provides researchers with general information useful for further insights into the rules of mRNA decoding, i.e., allows defining the pools of transcripts with different translation efficiency and to find regulatory contexts of mRNAs or their combinations important for translation efficiency using computational analysis (this will be considered below in more detail). According to the available experimental data, polysome profiling is, as a rule, applicable to the search for actively-translated mRNAs and the subsequent analysis, although the understanding of the mechanisms associated with the repression of translation in a certain pool of transcripts is of the same importance; perhaps, researchers will focus on this area in future. It should be also emphasized that most studies utilizing polysome profiling performed so far, involve the plants with annotated genomes. However, the use of this method is not limited to the plant species with annotated genomes and can be extended to other plant species, including those genomes that have not been yet determined or those already sequenced but poorly annotated.

### 2.2. Translating Ribosome Affinity Purification (TRAP) and TRAP-Seq

The experimental approach referred to as translating ribosome affinity purification (TRAP) is a modification of the traditional polysome profiling procedure and was for the first time described for *A. thaliana* [12]. This method utilizes the plant transgenic lines that express an epitope-tagged variant of ribosomal protein L18 (usually referred to as RPL18). As a rule, these plant transgenic lines express FLAG epitope-tagged RPL18 in the N-terminal region [12,13]. The cell lysates of transgenic plants are produced under the conditions that stabilize the ribosomes on RNA and block translation. The transcripts bound to the ribosomes that carry the labeled RPL18 are selectively separated using the absorption on anti-FLAG-M2 agarose. This enables ribosome capture from crude cell extracts by a single-stage immune precipitation (Figure 2) and, as a rule, allows the pool of RNAs (designated as TRAP RNA) that are actively translated to be obtained.

This method is described and discussed in detail in several papers [3,13,14,15]. Note that both the traditional polysome profiling approach and TRAP give analogous proportions of the small and large polysomes (i.e., ribosome profiles) [13]. Both approaches also have similar limitations on their application, namely, in the assessment of the number of ribosomes per mRNA length and the distribution of ribosomes along the transcript (see above). However, note that a wide use of the experimental TRAP approach is also limited by the available plant transgenic lines but, nonetheless, the use of transgenic lines gives certain advantages as compared with the traditional polysome profiling. This advantage consists in the possibility of not only constitutive, but also tissue-specific RPL18 expression by using different tissue-specific promoters [14]. Thanks to the tissue-specific RPL18 expression, TRAP is applicable to profiling of actively-translated RNAs in different populations of plant cells, namely, in (i) different root cells (epidermis, cortex, or endodermis); (ii) companion phloem cells, meristem cells, and leaf mesophyll cells; and (iii) microspores, pollen, and other plant tissues and cell types [14]. For example, the use of APETALA1, APETALA3, and AGAMOUS for expression of FLAG-RPL18 in early flower development allowed for the discovery of new levels of the expression control in developing flowers associated with differential translation [16]. A systemic analysis of the mRNAs in different specimens relative to the pollen grains within buds and in vitro-germinated pollen tubes has been performed with the help of the *A. thaliana* transgenic lines expressing epitope-tagged RPL18 under the control of ProLAT52 promoter, which allowed for the identification of a cohort of the transcripts that regulate late stages of pollination in flowering plants; this paves the way for better understanding of the pollen-based mechanisms that promote fertilization [15]. It should be emphasized that the in vivo proteomic studies of pollen tubes are extremely complicated because of the difficulties with pollen collection; the selective immune purification of the transcripts associated with the polysomes in pollen tubes in this case assisted in identification of the genes important for the in vivo pollen biology. Thus, the TRAP approach has an important advantage for efficient isolation of the population of mRNA complexes from particular cell types.

The sensitive moment when using TRAP approach is during the selection of the transgenic line that expresses FLAG-RPL18, which is extremely important for a successful analysis of the tissue-specific responses. A position effect associated with the T-DNA integration site in the genome of transgenic plants is known. In this regard, the new transgenic lines intended for this research should be selected bearing in mind the presence of known tissue-specific genes in the corresponding tissues or cell types. This will ensure selection of the most appropriate line for further analysis.

According to the current opinion, not only stable plant transformants, but also a transient expression of FLAG epitope-tagged RPL18 can be used for identification of the differentially-translated mRNA pools in plant genomes, for example, utilizing the agroinfiltration of *Medicago truncatula* hairy root cultures or of *N. benthamiana* leaves by *Agrobacterium rhizogenes*.

The FLAG tag may be also added to other proteins in order to determine their role in translation. For example, the expression of tagged oligouridylate binding protein 1 (UBP1) with subsequent immune purification of the mRNA–protein complexes (mRNPs) clarified the role of this protein in the dynamic and reversible aggregation of translationally repressed mRNAs in hypoxia [17]. In particular, UBP1 constitutively binds a subpopulation of the mRNAs with the 3’UTRs enriched for uracil under normoxic conditions. In hypoxia, UBP1 is associated with non-uracil-rich mRNAs, which increases its aggregation in microscopically-visible cytoplasmic foci, referred to as UBP1 stress granules (SGs). This UBP1–mRNA association leads to a global decrease in the protein synthesis. The translation limitation for the transcripts associated into SGs reduces the energy spending, thereby determining the priority in synthesis of the proteins that enhance plant survival in stress. The UBP1 SGs rapidly disaggregate during reoxygenation, which coincides with the mRNA return to polysomes. In this process, the mRNAs that are significantly induced and translated in hypoxia to a considerable degree manage to avoid UBP1 sequestration. Thus, it has been shown that the SG-nucleating RNA-binding UBP1 is a component of the mechanism that post-translationally reprograms plant gene expression, thereby enhancing plant survival in hypoxia [17].

### 2.3. Ribosome Profiling, or Ribo-Seq

Ribosome Profiling (RP), or Ribo-Seq, elaborated by Ingolia, Newman, and Weissman in 2009 [18], is based on the isolation and sequencing of the mRNA fragments protected by ribosome. This gives a “snapshot” of the ribosome positions along mRNA on a genome-wide scale, i.e., gives the possibility to determine both the number and positions of the ribosomes in the mRNA coding region in vivo (Figure 3).

As a rule, many studies use the RP experimental protocol, which comprises five interrelated stages: (i) preparation of RNA specimens with the arrested ribosomes; (ii) controlled hydrolysis of these specimens by RNase to generate small RNA fragments associated with a ribosome (referred to as footprints); (iii) their subsequent isolation; (iv) preparation of purified footprints with a size of 28–30 nucleotides; and (v) construction of the library and its high-throughput sequencing, as a rule, with the help of short-read sequencers. The deep sequencing reads of the footprints are analyzed using bioinformatics methods and the translation efficiency is derived by normalizing the number of reads of the footprints to the number of reads of the total transcriptome by RNA-Seq.

As is mentioned above, the first experimental protocol for ribosome profiling was described in 2009 [18] and has been constantly developed, in particular, for its application to different organisms [18,19], including plants [20,21] and plant organelles—chloroplasts [20] and mitochondria [22]. The individual protocols differ in the particular details providing optimization of each of the five interrelated stages, including the differences in tissue and cell processing; pH and composition of the buffer for cell lysis; prepurification of polysomes before RNase hydrolysis (done or omitted); type of RNase used for generating monosomes [23]; and the methods used to purify the monosome fractions and construct sequencing libraries. Ribosome affinity purification (TRAP method), including the tissue-specific purification, can be also used as the starting point for ribosome profiling [20].

In general, the RP results allow for determination of the precise positions of the translating ribosomes on mRNA with an unprecedented resolution, to a single nucleotide. The specialized software for analysis, interpretation, and visualization of RP data is currently available (for detailed review, see [24]). By assessing the relative number and location of ribosomes on mRNA, the researcher can estimate the general translation pattern i.e., to assess the translation efficiency, which is calculated as the ratio of translation (the data on the number of footprint reads in individual mRNA) to transcription (RNA-Seq data at the level of individual mRNA) (Figure 4). Note that it is possible not only to directly quantify the mRNAs that will be translated into proteins, but also to detect the new types of contexts in the plant mRNAs associated with translation, for example, uORFs (upstream ORFs) and frameshifts; to precisely determine the translation initiation site (TIS) of the main ORF; and to find new translated ORFs, including those residing in intergenic RNAs or putative noncoding short RNAs (ncRNAs) (Figure 4) [3,24,25]. The researcher gets these additional options thanks to the fact that the 80S ribosomes associate only with the portion of the transcript that will be most likely decoded into the protein product. The 80S ribosome and transcript will associate not only in CDS, but also in 5’UTRs if they contain an uORF, i.e., short translated reading frame located upstream of the main ORF (CDS), which may have an important role in translation regulation. Another most important aspect that can be studied in terms of the RP experimental data is assessment of the dynamics of ribosome movement along individual mRNAs and the rate at which certain codons are translated. This is possible because three nucleotide bases in the sequenced footprints are reflected in a periodic mode as a consequence of the ribosome movement along the mRNA coding region, since the ribosome moves along the overall coding sequence in a codon-wise manner, the 5’ region of ribosome footprints tend to be mapped at the same position of each codon.

Find below several examples which in our view illustrate the distinctive capabilities of RP in clarification of the fine mechanisms underlying the translational control in plants, such as the detection of new ORFs, including those annotated as noncoding RNAs and pseudogenes. In particular, the study of translation regulation under normoxic and sublethal hypoxic conditions (hypoxia) in *A. thaliana* shoots with the help of RP not only detected an inhibitory effect of the uORF on the translation of downstream protein coding regions in normoxia, which was further modulated by hypoxia, but also determined the alternatively spliced mRNAs as well as the fact that ribosomes were associated with certain noncoding RNAs [21]. An RP study of the maize shoots under drought showed a statistically significant change in the translation efficiency of 931 genes, which according to further analysis of the transcripts was associated with the nucleotide composition of the sequence, including GC content, length of coding sequences, and normalized minimum free energy. In addition, potential translation of 3036 open reading frames (uORFs) in 2558 genes was detected; the authors believe that these uORFs are able to influence the translation efficiency of the downstream main open reading frames (ORFs) [26]. In another study, the Ribo-Seq data detected 27 and 37 translated sORFs (short ORFs) among the annotated noncoding ncRNAs and pseudogenes of *A. thaliana*, respectively [27]. Moreover, 187 translated uORFs were identified with a high degree of reliability. In addition, the events of translation from the start codons other than AUG were identified in the dataset among both annotated genes and uORFs. They also demonstrated that 15 of the 19 detected single-exon sORFs had homologs in various flowering plants, which suggests their functional significance [27].

Lukoszek et al. [28] used RNA-Seq and Ribo-Seq to assess reprogramming of the *A. thaliana* global gene expression during a long-term heat shock (3 h at 37 °C) at both the transcriptional and translational levels. They have shown that translation is globally impaired in the early period of the heat impact (15 to 45 min), while the stress response appears mainly at the expense of transcriptional programs. In this process, a long-term stress impact (3 h) activated translational programs, which eventually form the adaptive response. The transcripts regulated via translation display a number of common characteristics, namely, the presence of relatively conserved A/G-rich motifs in their 5’UTRs or 3’UTRs that are similar to the sequences identified as protein-binding nucleotide motifs. Another specific feature widespread among the genes upregulated in heat stress is that they are less inclined to form secondary structures, which is likely to ensure their binding with ribosomes and to enhance translation. In addition, several transcripts prevalently induced by heat contain a putative G2 quadruplex in their 5’UTRs. Note that an increased number of reads for RP footprints in quadruplex structures correlates with an expanded expression of the downstream CDSs. This suggests an important role of these structures in translation activation of the downstream ORF according to yet unknown mechanism [28]. Ribosome profiling has been used to analyze translation of the chloroplast transcripts in maize shoots in response to changes in light conditions. According to the experimental data, all chloroplast mRNAs except for psbA maintain similar numbers of ribosomes after short-term changes in light conditions but nonetheless are more efficiently translated in the light. On the other hand, the psbA mRNA displays a sharp increase in the ribosomes over several minutes after the plants are transferred to light and restores a low ribosome loading during 1 h in the dark, which correlates with the need to replace the damaged psbA in photosystem II. These results emphasize the unique translational response of psbA in mature chloroplasts, indicate the particular light-regulated steps in the context of photosystem II activity maintenance, and provide the background for the study into the mechanisms underlying both the psbA-specific and genome-wide effects of the light on the translation in chloroplasts [29].

The RP technology was also used to study several aspects in the translation of *A. thaliana* mitochondrial genome in a dynamic mode. As has been shown, the mitochondrial mRNAs are differentially-translated; in this process, the translational levels of the transcripts encoding the subunits of mitochondrial protein complexes, in particular, complex V, proportionally correlate with the stoichiometry of respiratory chain subunits. In general, the mitochondrial translation is shown to be controlled at the level of individual mRNAs and is directly involved in the activity regulation of plant mitochondria [22].

Note that Ribo-Seq technology is currently at a relatively early stage of its development, which leads to some experimental difficulties and technical artifacts influencing the Ribo-Seq data interpretation [18]. In particular, the RP results may display statistically significant differences associated with the modifications of one of the five stages in the basic protocol, such as the conditions of cell lysis, composition of buffer solutions, selection of nucleases and the absence of pronounced specificity to the sequences to be cleaved, and construction of the library; even more so as these details in many cases are not analyzed in a systematic manner [19,23]. This suggests the need to systematically study the effects of the corresponding experimental parameters of the used RP protocols [19].

The RP technique also has its limitations. According to the current scientific consensus, the basic limitation of RP approach is a static position of ribosomes along the mRNA. This prevents distinguishing between the ribosomes involved in translation from the ribosome in a steady-state [19]. Thus, the methods used in the majority of studies involving RP can overestimate the translation efficiency because of the data related to monosomes, in which mRNA is also protected by a ribosome (Figure 5A) [18,23,26]. Underrepresentation of the transcript regions with ribosome stacking is also possible; this is associated with the stacked polysomes and may prevent hydrolysis in monosomes, because of inaccessibility to RNases (Figure 5B) [4,30]. Correspondingly, the recommendation for an additional direct measurement of the polysome-protected mRNA looks most reasonable to overcome this limitation of the RP approach [26].

Note that the Ribo-Seq technique now is mainly applicable to study translation of the organisms with annotated genomes since the deep sequencing data for footprints are represented by very short reads (28–30 nucleotides), which, as a rule, are analyzed by mapping onto the genome data (Figure 4).

However, the current limitations of this method can be bypassed and this experimental technology will remain a useful tool in the omics [30]. RP data with high resolution is a priceless resource for studying noncanonical start codons and alternative start sites and can be useful for characterizing translation of different isoforms of transcripts, identifying new translated ORFs and their quantifying, and, in general, for improving the genome annotation of poorer characterized organisms. Additional ribosome profiling can also be a proxy for the proteome or assist in proteomics studies [27,30].

Completing this section, note that the genome-wide profiling of the transcripts associated with ribosomes utilizing one of the above experimental approaches may highlight the new aspects in gene expression unvisualizable by an ordinary profiling of the total cellular mRNA. In Table 1, we attempted to consolidate the advantages, limitations, and areas of applicability of the discussed experimental approaches to the study of differential translation on a genome-wide scale. Undoubtedly, selection of the appropriate experimental approach depends on the particular aims set by a researcher.

## 3. Computational Algorithms for Predicting the Features of Plant mRNAs Important for Differential Translation

The above described experimental approaches make it possible to detect the specific pools of transcripts with characteristic differential translation. Several computational resources are useful for identification of regions of specific structural features in mRNA nucleotide composition that can mediate differential translational control.

In this section, we summarized the resources and some computational algorithms that have been used to form the samples of target plant transcript sequences and to predict their peculiar characteristics, as well as their main functions and domains of application. Note that the resources and the corresponding software are rather numerous and, in fact, require a separate review. Here, we consider only those that have given the data on and predictions of regulatory contexts in transcripts with further experimental confirmation.

### 3.1. Preparing Datasets for Analysis

The key preparatory stage in the in silico predictions is a construction of the most representative sets of sequences for the transcript pools differing in their translation efficiency. Note that the researcher needs not only full-sized transcript sequences (cDNA), but also the sequences of individual regions of these transcripts, namely, coding (CDS) and untranslated (5’UTR and 3’UTR) regions, which, as mentioned above, can also contribute to translation efficiency. Currently, many internet resources have been elaborated that allow sets of such sequences to be downloaded, including the sequences for plants. In particular, TAIR is the information source for the model plant *A. thaliana* (https://www.arabidopsis.org/download/index-auto.jsp?dir=%2Fdownload_files%2FSequences%2FTAIR10_blastsets) [31], which is widely used for loading 5’UTR, CDS, 3’UTR, and cDNA sequences using the tools “Download”, “Sequences”, and TAIR10 blastsets [32,33,34]. Another information resource containing CDS, cDNA, 5’UTR, and 3’UTR sequences of the representatives of six key kingdoms of the living organisms, including plants, is JetGene. JetGene is publicly available at http://jetgene.bioset.org/; its data are stored and updated at the Ensembl server [35]. The intuitively clear and friendly JetGene interface allows the cDNA, 5’UTR, CDS, and 3’UTR sequences to be extracted in a FASTA format, including the specific samples on user request. Note that only the sequences with complete information about the full-sized transcripts are in most cases selected for further analysis.

Once the sets of sequences (5’UTR, CDS, and 3’UTR) for the pools of differentially-translated transcripts are obtained, the researcher has to select for analysis the regions of transcripts and regulatory sequences that may be potentially involved in translation modulation. According to the current opinion, the complex multilevel information is encoded in the full-sized mRNA sequence (transcript) in general and in its individual parts—5’UTR, CDS, and 3’UTR (Figure 6). This gives researchers the grounds to include all these regions into in silico analysis to characterize the differentially-translated plant transcripts. Note that translation initiation is, as a rule, the stage limiting the translation rate and 5’UTR plays here the decisive role. The length, nucleotide composition, secondary structures, and regulatory elements of a smaller size, such as upstream start codons (uAUGs), uORFs, nucleotide motifs, and several other features in the 5’UTRs of transcripts, are closely examined in terms of their contributions to the translation efficiency. In this process, the probability to find the potential regulatory regions and contexts and to clarify how their properties influence the translation efficiency will be higher if more traits of this kind are involved in the initial in silico analysis.

The further aims of the researcher could be (i) to assess the variations in distribution of individual traits in the sequences from the examined transcript pools and to figure out the statistically significant differences that are positively correlated with the translation efficiency; (ii) to find and determine the statistically significant representation of the potential regulatory contexts in the transcripts with different translation efficiencies; and (iii) to identify the specific regulatory sequences if they are present in the examined pools.

### 3.2. Statistical Methods

The methods of mathematical statistics have been rather efficiently used for solution of the first task. As a rule, basic and extended statistical analyses are performable with the help of the available standard programs, such as Excel, STATA, and IBM SPSS Statistics 20 [26,32,33,34]. For example, the genome-wide monitoring of the changes in the translation efficiency of individual mRNAs in *A. thaliana* shoots after heat shock have demonstrated translation repression for the majority of mRNAs; however, some mRNAs still followed the differential translation pattern. Analysis of the differentially-translated mRNA sequences demonstrated that only some characteristics, such as the G + C content in 5’UTR and cDNA length, are putatively involved in the mechanisms providing discrimination of the mRNA loading with ribosomes and are associated with differential translation of a certain transcript cohort in response to a high temperature. In particular, the translationally active mRNAs have a low G + C content (on the average, 36%) versus the transcripts with repressed translation (42%). This selection mechanism also influences the differential polysomal loading of the transcripts associated with stress and, as a consequence, the efficiency of their translation [32].

In general, the methods of mathematical statistics have made it possible to (i) find the characteristics that are representative for the analyzed mRNA, (ii) discard the characteristics the effect of which can result from a bias to the group of particular genes, and (iii) determine the statistically significant differences displaying a positive correlation with the relative translation efficiency.

### 3.3. Methods for Identification of RNA Motifs

The methods that are used to identify potential regulatory motifs in mRNA sequences assess the statistical significance of represented potential regulatory motifs in the mRNAs with different translation efficiencies. These methods are in general the same or very similar to those for DNA motifs except for the methods addressing the DNA conformational properties. The latter utilize physical parameters of DNA double helix and can be plied both to prokaryotic [36] and eukaryotic [37] genomes. Having appropriate parameters to convert letter representation of RNA into numerical representation, the same methodology could be applied to analysis of mRNA. Conventional approaches are based on accounting for conserved nucleotides within a short motif. One of the most frequently used programs for the detection of motifs in the transcript pools with different translation efficiencies is МЕМЕ, which is based on the maximal likelihood optimization [38]. Ease of use and a wide set of the accompanying programs for visualization and further search are advantages of this program. The MEME suite comprises four main sections, namely, motif discovery, motif enrichment analysis, motif search, and motif comparison, altogether 14 different tools. This toolkit allows the researcher to both determine motifs de novo and to scan a dataset of sequences for the matches of the already known motifs. MEME shows a schematic arrangement of the found motifs on the initial sequence, constructs a graphical representation for them, and computes statistical significance (*p*-value) for these motifs.

In particular, MEME suite has allowed identification of a nine-nucleotide-long element present in both the 5’UTRs and 3’UTRs of numerous *A. thaliana* and *Gynandropsis gynandra* transcripts; the authors named it MEM2. Later, it was experimentally confirmed that the MEM2 motif residing in the 5’UTR was necessary for preferential protein accumulation in the mesophyll cells. It is assumed that this motif can be involved in the mechanism guiding preferential cellular accumulation of several enzymes necessary for C4 photosynthesis, which provides a more efficient carbon capture as compared with the ancestral C3 pathway [39]. The MEME suite has been used in a comparative analysis of the 5’UTR sequences for steady-state and polysomal *A. thaliana* mRNAs and allowed for discovery of two motifs (TAGGGTTT and AAAACCCT) present in many genes, which potentially suggests their contribution to the translation efficiency. Furthermore, it has been experimentally shown that only one of these motifs, TAGGGTTT, regulates gene expression at the level of translation [33].

However, the search for the motifs using this tool also has some limitations. Among the serious disadvantages of this program is the trend to find very long motifs (over 20 nucleotides), these motifs are present only on a small subgroup of sequences and/or frequently repeated motifs in one or just a few sequences. Although statistical significance (*p*-value) of such motifs is very high, the motifs themselves, as a rule, are rarely of any biological/practical interest and represent statistical artifacts.

Most likely, these limitations are the main reason why several studies of motifs failed to bring any positive results [8,32,33]. Correspondingly, other computational approaches were used for this search and their statistically significant representation in the transcripts with different translation efficiency, for example, by comparing the frequencies of mono-, di-, and trinucleotide sequences. Statistical tests, for example, the Kolmogorov–Smirnov or Fisher test, allow the detection of statistically significant differences in such nucleotide distributions. Moreover, the use of linear or logistic regression makes it possible to detect not only the individual contribution of each sequence, but also the effect of their combinations. In particular, partial least regression analysis has been applied to the detection of the short regions residing in the neighborhood of the 5’-proximal region of 5’UTR that can play an important role in differential translation in response to heat shock [8]. However, the linear or logistic regression methods are also not free from limitations. For example, it is not practically feasible to analyze motifs with a length of four nucleotides or longer, because their frequencies sharply decrease and, as a consequence, the computation of statistical characteristics becomes too complicated. In addition, these methods do not take into account the locations of motifs on sequences, which in terms of biology mean the equal contributions of the codons residing far from the translation start codon and in the immediate proximity.

### 3.4. Detection of Other Context Features of RNA

Statistical approaches are rather efficient when a set of potential characteristic features is determined for a pool of sequences and is used as a reference in the analysis. However, the specific cis-regulatory sequences in mRNAs that can modulate translation are identified using specialized approaches and/or resources for their prediction. The examples below illustrate the approaches to predict cis-regulatory sequences in the case studies of internal ribosome entry sites (IRESs) and upstream ORFs (uORFs), first and foremost, conserved peptide uORFs (CPuORFs).

IRES are the nucleotide sequences that mediate translation initiation of alternative reading frames (aORFs) under stress conditions, when the trivial cap-dependent translation mechanism is inhibited without the corresponding changes in gene transcription [40]. In general, the IRESs of plant mRNAs, unlike the IRESs of viruses, display considerable diversity in both nucleotide composition and structure [41]. Despite this diversity, characteristic functional modules are distinguishable in the IRESs, namely, (i) the presence of several start codons and their localization and (ii) the fact that some IRESs carry short conserved modules, which are recognized by the plant translational machinery and are directly involved in the immobilization of ribosome small subunit [42]. Polypurine blocks residing close to the start codon, which may be directly involved in the immobilization of ribosome small subunit, are an example of such conserved motifs [43].

The mRNAs potentially carrying IRESs can be selected by analyzing the experimental data obtained by polysome or ribosome profiling followed by deep sequencing and/or by mass spectrometry analysis. First and foremost, such mRNAs must retain a high level of their translational activity under the impact of adverse environmental factors and carry additional alternative start codons. The following strategy is appropriate for further selection and analysis of the mRNAs carrying IRESs. (i) Interspecific comparison of the transcript sequences of homologous genes, which allows for identification of the conserved region in the vicinity (30–50 nucleotides) of the alternative start codon followed by (ii) assessment of the context of the alternative start codon, the optimal neighborhood of which may suggest that translation can be potentially started from it. This strategy has been successfully implemented for predicting translation initiation of a short aORF with involvement of a polypurine block via internal ribosome entry [43]. Note that a commonly accepted confirmation for an IRES activity is still its ability to provide a coordinated translation of reporter genes within a bicistronic transcript (see below).

The advent of RP and high-throughput sequencing technologies made it possible to determine the translation start sites and to discover numerous mRNAs with aORFs that (i) may have a putatively inert sequence that acts as a mere translation barrier upstream of the main ORF or (ii) may encode short peptides referred to as CPuORFs [44]. The main difference between CPuORFs and the other ORFs is their length and, although there are no strictly defined frames, the ORFs shorter than 200–250 codons are regarded as short. In general, the search for CPuORFs is analogous to the approach for prediction of main ORFs and the strategy utilizing interspecific comparison of CPuORF sequences to identify the conserved regions is in most cases used for this search and estimation of the coding potential. This strategy is based on revealing the homology between such short peptide sequences and to a considerable degree depends on the range of the species selected for comparison. For example, it is quite possible that the fact of preservation of CPuORFs within the plant species selected for comparison is insufficient to reveal the conserved regions. In this case, the analyzed CPuORFs will not be identified although their sequence is sufficiently conserved among the other species. When comparing the CPuORF sequences among closely related species, the observed similarity between short uORF peptide sequences may result from nucleotide sequence retention rather than conservation of these peptides. In order to overcome the problems associated with selection of the species for comparative analysis, a new method for CPuORF identification, BAIUCAS, was developed and tested [45]. BAIUCAS utilizes sets of EST (expressed sequence tag) data for thousands of plant species to search for homologs. The BAIUCAS algorithm consists of six successive stages: (i) exhaustive search for uORFs; (ii) search for homologs of CPuORF amino acid sequences over EST databases using tBLASTn; (iii) selection of CPuORFs based on the conservation of the stop codon position; (iv) selection of the CPuORFs conserved in a wide range of species, i.e., the CPuORFs with the conserved stop codons detected in each of several taxonomic categories; and (v) and (vi) are filtration stages, which excludes the “false” conserved CPuORFs [45]. Using this approach to the search for CPuORFs, 16 *A. thaliana* CPuORFs were identified; five of them are the new CPuORFs involved in the translation regulation of the main ORF, which has been experimentally confirmed [46].

The list of the computational approaches that have been so far successfully used in decoding the specific structural features in nucleotide composition of the plant mRNAs that mediate differential translation control is rather short. One of the possible efficient computational tools for analyzing the translatomes and predicting numerous regulatory codes in transcript sequences could be artificial neural networks (ANNs). This assumption relies on the facts that (i) most neural network architectures is theoretically able to approximate any function, i.e., it is potentially possible using ANNs to construct a model for almost any biological pattern, and (ii) the capabilities of the supercomputers have reached the appropriate level to model biological processes using neural networks. However, a positive result of analysis depends first on an adequately selected architecture of the network and second on the amount and composition of the training sample and a training strategy. Several recent reports confirm the utility of ANNs in deciphering the molecular mechanisms involved in decoding the eukaryotic genome. For example, the ANNs constructed based on RP data have been used to predict the yield of protein products [47]; to search for the motifs potentially able to influence translation [48]; to extract the biologically important information from omics data [49]; and to simulate the interaction between nucleic acids and different types of ligands (protein and peptides) [50]. The ANN potential is broad enough and it can be expected its broad application to the research in diverse and multilevel mechanism underlying translation in plants.

## 4. Approaches for Experimental Verification of the Systemic Experimental Data and Theoretical Predictions of Intrinsic Features of Plant mRNAs Important for Differential Translation

As a rule, the experimental results on determination of the pools of differentially-translated plant mRNAs as well as the predictions of the regions and nucleotide contexts in differentially-translated transcripts require experimental confirmation of their functionality and contribution to the translation efficiency. In this section, we will consider the main experimental approaches that allow for convincing confirmation of which particular regions in individual mRNAs and/or features within plant transcripts are important for modulation of translation. It should be emphasized that methods for high-throughput experimental verification, i.e., concurrent analysis of a large pool of regulatory regions, are yet not available; thus, potential regulatory sequences are examined for each individual transcript [4].

### 4.1. Direct Measurments of Individual Transcripts

Several tools for determining the changes of individual transcripts at the level of translation are available, including (i) the systems for in vitro translation based on measuring the incorporation of labeled amino acids, such as FUNCAT (fluorescent noncanonical amino acid tagging), SILAC (stable isotope labeling by amino acids in cell culture), BONCAT (bioorthogonal noncanonical amino acid tagging), QuaNCAT (quantitative noncanonical amino acid tagging), and PUNCH-P (puromycin-associated nascent chain proteomics), or cell-free protein expression systems, such as the wheat-germ extract, which contain all factors necessary for translation of the target transcript; (ii) toeprinting, or the primer extension inhibition assay, utilizing reverse transcription to study the interaction of ribosomes with the target transcript; (iii) enzyme immunoassays, in particular, Western blot hybridization; and (iv) mass spectrometry-based methods for identifying the changes in the proteome or their combinations with in vitro translation methods, for example, PUNCH-P. The review by Mazzoni-Putman et al. [4] describes the principles, advantages, and limitations of these methods in detail.

However, these research methods are in most cases suitable for assessing the general changes in translation, require considerable time, amount of reagents, and specialized equipment; correspondingly, most research works now use the strategy of reporter systems for studying the structure–function characteristics of the target sequences. The strategy of reporter genes considerably enhances such research since it is much easier to record the protein product of a reporter gene as compared with a studied gene. It should be also emphasized that the reporter genes code for the proteins that display either unique specific features or unique enzyme activities, allowing their products to be easily isolated from the totality of intracellular and extracellular proteins. Thanks to these advantages of reporter systems over the other methods for studying the regulation of gene expression, they have been widely used for experimental verification of the regions and nucleotide contexts in differentially-translated transcripts. For studies of this type, expression cassettes are constructed that carry the reporter gene sequence with the expression controlled by a particular regulatory region or sequence selected by researcher (Figure 7). Researchers have at their disposal several reporter systems that have proved their efficiency in the studies of potential regulatory sequences or the nucleotide contexts that modulate translation in plant systems, in particular, β-glucuronidase (GUS); different variants of fluorescent proteins (for example, GFP and RFP); luciferases (Renilla luciferase, RLuc, and firefly luciferase, FLuc); and thermostable lichenase (LicBM) [51,52]. Commercial substrates and kits as well as the quantification methods for assessing the corresponding protein products are available for these reporter systems. The main approaches that have been applied in the studies of various regulatory regions or nucleotide contexts within transcripts with the use of reporter systems are illustrated below.

When experimentally confirming the role of the full-sized 5’UTRs in transcripts, these sequences are cloned upstream of the 5’ region of a reporter gene (Figure 7). Quantitative estimate of the reporter gene protein product when using different target 5’UTRs versus the known translational enhancers of various origins makes it possible to assess their contribution to the translation efficiency [51]. For example, *A. thaliana* mRNAs that are stably translated under any growth and environmental conditions have been found by polysome profiling. Testing of the translation capability of mRNA 5’UTRs of candidate genes using the reporter gene strategy has convincingly demonstrated that the 5’UTR of 47 cold-regulated genes are an efficient translational enhancers, which enables a stable high level translation under any conditions of plant growth. This suggests the utility of this method for plant biotechnology, for example, when engineering plants producing biologically active substances or the plants resistant to some stress factors, including the schemes that involve genome editing technologies [9].

Recombinant 5’UTR sequences are designed for identification of the cis-regulatory elements in these mRNA regions, first and foremost, the motifs or specific nucleotide contexts potentially able to influence differential translation, using for this purpose a (i) combinatorial approach, (ii) site-specific mutagenesis, (iii) translation assessment of the second (3’-terminal) ORF of a bicistronic transcript, (iv) deletion analysis, (v) frameshift mutations, or a combination of these methods (Figure 7).

### 4.2. Site-Specific Substitutions

The combinatorial approach utilizes the fact that mutual substitutions of the predicted regulatory motifs are introduced into the pairs of 5’UTRs with the same lengths but with experimentally confirmed significant difference in translation efficiency (Figure 7a). Matsuura et al. [8] successfully used this approach to identify new cis-regulatory elements in 5’UTRs that influence the differential translation of *A. thaliana* transcripts in response to heat shock (HS). The genome-wide analysis of the changes in polysome loading of the transcripts in Arabidopsis cell culture allowed for selection of a set of genes with different translational responses to HS. The 5’UTR nucleotide sequences of the transcripts that change the level of reporter protein in the protoplasts affected by HS were used to predict the regulatory elements in 5’UTRs with the help of partial least square (PLS) method. These computational predictions suggested that two short regions residing in the vicinity of 5’-proximal region of the 5’UTR can play an important role in the relative activity of reporter protein and, thus, may be regarded as cis-regulatory region candidates. In order to experimentally confirm the predictions on the importance of these 5’UTR regions in differential translation control, a series of mutual substitutions of these regions in the pairs of 5’UTRs with equal length but different translation efficiencies were analyzed. Analysis of the reporter gives convincing evidence that the 5’-proximal region of the 5’UTR plays a key role and that certain specific determinants in 5’UTR mediate the differential translation in response to HS [8].

Site-specific mutagenesis makes it possible to introduce substitutions of one or a group of nucleotides within strictly defined regions of nucleic acid sequences. Comparison of the levels of a reporter protein translated when controlled by a native regulatory region (for example, 5’UTR) and the same regulatory regions but with the introduced mutations (Figure 7b) demonstrates how the modification of the primary sequences of nucleotide contexts and/or their secondary structures modulates the translation of specific mRNAs. This approach, along with other molecular methods (Western blotting, qPCR, polysome fractionation, and so on), has emerged to be most efficient when studying both the mechanism underlying formation of RNA G-quadruplex in the 5’UTR of the SUPPRESSOR OF MAX2 1-LIKE4/5 (SMXL4/5) mRNAs and the clarification of the role of a specialized structure, the regulator of phloem formation. In particular, a novel zinc finger protein, JUL, was identified; it specifically bound to consecutive guanine repeats in the 5’UTR of SMXL4/5 and induced RNA G-quadruplex. Moreover, convincing experimental data that both JUL1 and G-quadruplex are necessary for strong translation suppression rather than a single-stranded G-rich element have been obtained using the strategy of reporter systems. This suggests that the suppression of translation is caused by either JUL1-mediated formation of G-quadruplex or the G-quadruplex/JUL1 complex recruits an unknown translational suppressor [53].

Site-specific mutagenesis has emerged to be efficient for clarifying the role of the TAGGGTTT motif, overrepresented in the 5’UTRs of the transcripts regulated at the level of translation. A comparative study of two constructs, one with a native 5’UTR carrying this motif and the other with the 5’UTRs carrying mutations in this motif, has shown that the transcripts with the native 5’UTR are more efficiently translated provided that the number of transcript are equal. Thus, it is experimentally proved using reporter genes that the TAGGGTTT cis-element regulates expression of the gene particularly at the level of translation [33].

### 4.3. Analysis Using Deletions

Deletion analysis of 5’UTRs, implying construction of truncated variants of sequences and their subsequent fusion with a reporter gene, makes it possible to identify nucleotide contexts decisive for maintaining the structure of RNA that can be in two particular conformations, as for example, in riboswitches (Figure 7c). Note that an important specific feature of the riboswitches is their ability to both activate and inhibit translation from the controlled ORF, due to the presence of a specific regulatory region, the aptamer. This region with a particular secondary and sometimes tertiary structure, has the properties of a receptor for small molecules (ligands). In the overwhelming majority of cases, riboswitches reside in 5’UTRs. A deletion analysis of the *A. thaliana* phytoene synthase (PSY) 5’UTR has shown that the long 5’UTR variants (403 and 330 nucleotides) with two predicted sequences of convertible RNA conformations, similar to riboswitches, inhibit translation of the reporter gene, in contrast to the short variant (252 nucleotides) of the PSY 5’UTR, lacking such hairpin structure. This allows the short 5’UTR variant to pass the translational control and rapidly elevate the protein levels [54].

### 4.4. Translation from Alternative ORFs

Translation of the alternative ORFs may be provided by a specific mRNA region, referred to as IRES (internal ribosome entry site). The functionality of an IRES can be experimentally confirmed by constructing bicistronic transcripts (Figure 7e) [40,41]. Note that the use of bicistronic constructs when studying the IRES functionality is determined, in particular, by the need to normalize a relative efficiency of the IRES-guided translation as compared with the cap-dependent translation. For example, functionality of the initially predicted IRES in the mRNA for a tobacco heat shock protein [55] was experimentally confirmed utilizing the ability of coordinated expression of reporter genes within a bicistronic transcript [56]. A structural analysis of the 5’UTR of a maize heat shock protein (Hsp101) mRNA has shown the presence of three stem loops towards the 5’ end, which suggested the functioning of the 5’UTR structure as an IRES. This assumption was experimentally confirmed by comparing bicistronic reporter constructs; in particular, it has been shown that the overall hsp101 5’UTR sequence (150 nucleotides) acts as an IRES, since the deletion of 17 nucleotides from the 5’ end decreases the translation efficiency of the reporter gene transcript by 87% as compared with the control sequence [57]. A functional analysis of the bicistronic constructs has revealed IRESs in the *A. thaliana* WUS mRNA, coding for the homeodomain transcription factor WUSCHEL (WUS), as well as the role of an additional protein, AtLa1 (an RNA-binding factor). As demonstrated, AtLa1 initiates an IRES-dependent WUS mRNA translation by binding 5’UTR (WUS is responsible for supporting the *A. thaliana* apical meristems under stress conditions) [58].

Alternative ORFs are among the most abundant regulatory elements in mRNAs; they are frequently present in the 5’ leader regions of eukaryotic mRNAs (designated uORFs). Such uORFs may negatively modulate the translation efficiency of the downstream main ORF. According to the current estimate, approximately 20% of the *A. thaliana* protein-coding genes contain uORFs in their mRNA 5’UTRs [59]. Initially, such regulatory sequences are either predicted via searching for the conserved peptide uORFs (CPuORFs) [45] or experimentally detected, in particular, by ribosome profiling during, for example, plant cell response to a stress [43,60,61]. The highest interest of researchers is associated with the function of the regulatory sequences, such as CPuORFs, which are able to act in a sequence-dependent manner or cause ribosomal arrest, thereby modulating translation of the main ORF.

### 4.5. Frameshift Mutations

The approach of frameshift mutations utilizes concurrent introduction of deletions and insertions at −1 and +1 positions; this procedure changes only the amino acid composition of a peptide sequence coded for in CPuORFs but retains the presence and unchanged length of the overlapping CPuORFs (Figure 7d). This method has been used to analyze 16 recently predicted conserved CPuORFs of *A. thaliana* for assessing a sequence-dependent effect of each CPuORF on expression of the main ORF. A comparative analysis of the reporter protein activity of the variants when the translation is controlled by either native CPuORFs or the CPuORFs with introduced frameshift mutations has identified five novel CPuORFs that repress the expression of the main ORF in a sequence-dependent manner. Moreover, it has been convincingly demonstrated that the C-terminal regions of four of these CPuORF-encoded peptides play a crucial role in repressing the translation of the main ORF [46]. The functionality of three *A. thaliana* CPuORFs in arresting ribosomes during translation was tested in another study. This mechanism of CPuORF action was clarified using toeprinting analysis and the additional experimental evidence was obtained by constructing the following three types of reporter constructs. (i) With the CPuORF initiation codon removed from each reporter construct of the native CPuORF by replacing AUG with AAG; (ii) with frameshift only mutations, introduced to the CPuORF sequences; and (iii) with both removed initiation codon and frameshift mutations in CPuORF sequences. A comparative testing of all types of reporter constructs has shown that removal of the initiation codon from CPuORFs considerably increases the reporter gene expression; the frameshift mutations in CPuORFs also efficiently increase the reporter gene expression, although to a lower degree as compared with the removal of initiation codon; while the simultaneous presence of frameshift mutations and absence of the initiation codon have almost no effect on the reporter gene expression. These results clearly demonstrate that (i) the peptide sequences are partially responsible for strong repressive effects of these CPuORFs on the main ORF expression; (ii) repression of the main ORF expression (in this case, the ORF of reporter transcript) depends on CPuORF translation; and (iii) these CPuORFs induce ribosomal arrest and thereby considerably inhibit expression of the main ORF [62]. Thus, it is possible not only to insert regulatory sequences that control the reporter gene translation into the constructs carrying this reporter, but also to introduce manifold modifications, which allows their functional role in a key biological process, translation, to be assessed.

### 4.6. Characteristics and Reasoning for Selection of a Verification Method(s)

It should be emphasized that the selection of a method for assembling reporter constructs is of highest importance, since it is necessary to clone the target regulatory sequences with a reporter gene without the introduction of additional nucleotides, which may influence the translation modulation. The classical restriction–ligation cloning method does not allow generation of such constructs and requires several cloning stages. Several more economical and efficient technologies facilitating and accelerating the design of such constructs have been recently proposed. These technologies make it possible to produce seamless fusions of a “regulatory sequence–reporter gene”. Most of them utilize the recombination between homologous sequences residing at the ends of the DNA fragments to be assembled [63]. For example, the Gateway® cloning system is based on the well-characterized site-specific recombinase system of the lambda phage for recombination of DNA segments. The DNA segments are flanked by the ATT recombination sites, which provide seamless assembly of almost any sequence [64]. However, this system also has certain limitations, namely, the need that the sequences overlap for at least 15 nucleotides at their ends. Correspondingly, the assembly of nonoverlapping DNA fragments requires additional terminal extensions or the use of bridge oligonucleotides [65]. Moreover, in our view, the approaches, such as Golden Gate system [66,67] or CPEC (circular polymerase extension cloning) strategy are more purposeful for designing the regulatory sequence–reporter gene constructs [68]. The Golden Gate method utilizes the IIS type restriction fragments to generate 4-nucleotide sticky ends, flanking each DNA portion, which then can be effectively ligated using T4 ligase. The assembly is performed in a single reaction and gives a seamless or nearly seamless target construct. This is determined by that the IIS type recognition sites are removed during ligation leaving only four nucleotide which positions may be determined by researcher [69,70].

The principle underlying the CPEC method utilizes the polymerase extension mechanism and one DNA polymerase for the in vitro assembly and cloning of sequences in any vector in a single-stage reaction. CPEC allows for an integrated, combinatorial, or multifragment assembly of sequences as well as for routine cloning procedures [68]. Thus, the Golden Gate and CPEC technologies have important advantages suggesting their utility in designing of the reporter constructs intended for studying and experimentally verifying the role of the regulatory regions in transcripts during translation.

As for the functional assessment of the constructs carrying a target regulatory sequence fused with a reporter gene, two basic experimental approaches are used: they utilized (i) a transient (temporary) and/or (ii) stable (constant) expression of reporter constructs in plants [4,71]. In the case of the transient expression of reporter constructs, transfection of the protoplasts derived from leaves of *N. tabacum* L. cv. BY2, lettuce, or *A. thaliana* are used as well as agroinfiltration of leaves (*N. benthamiana*, *N. excelsior*, *N. tabacum* var. xanth, or *A. thaliana*) or plant cell suspension culture (*N. tabacum* BY2 or *A. thaliana* T87) [4,71,72] (Appendix A). In the current view, the transient expression of reporter constructs is regarded as a more efficient approach because of lower material and time expenditures. The protocols for generation of protoplasts and agroinfiltration have been elaborated for many model and non-model plants, widening the range of the used plant objects. However, there are some limitations in the application of transient expression associated with the variation of protoplasts in the transformation efficiency and with the delivery of constructs to plant cells in agroinfiltration [4,73].

A stable expression of reporter constructs in plants requires production of transgenic plants or transgenic plant cell suspension cultures. This makes it possible to solve increasingly more complex problems of translational control, such as translation regulation under different growth and stress conditions or long-term physiological effects of certain changes in a sequence that modulate translation. In particular, polysome fractionation of control and transgenic plants makes it possible to confirm that the transcript of a reporter gene controlled by a tested regulatory sequence is actually associated with the polysome fraction. Thus, it is possible to assess the translational status of the mRNA of a reporter gene fused with the tested regulatory sequence, including under the effect of stress factors [8].

When using the methods involving both stable and transient expressions, the choice of an adequate control is of a paramount importance to ascertain that the change in expression of the reporter protein is actually associated with the change in translation (rather than with transcription, protein stability, and so on) [4,51].

According to the available experimental data, the strategy of reporter systems is in demand for a wide range of studies into individual regulatory sequences or their nucleotide contexts. The use of this strategy gives the unique data on the functional role of target sequences in translation efficiency; however, this strategy is, as a rule, supplemented with other methods. The choice of a particular method depends on the regulatory sequence or its context to be studied be it a full-sized 5’UTR or its regulatory elements, such as RNA G-quadruplex, IRES, uORFs, and so on, which is in part summarized above and is comprehensively described in the corresponding publications.

## 5. Conclusions

Translation plays a key role in the overall implementation of genetic information and the new knowledge about the intricate and multilevel information encoded in the mRNA sequence are of a paramount importance. The research into translation has revealed many new and interesting facts about the structural and functional role of the mRNA regulatory sequences that mediate differential translation. In particular, this success has been determined by the use of state-of-the-art technologies for assessing the translational statuses of individual mRNA species on a genome-wide scale in combination with computational algorithms and the methods for experimental verification, summarized here (Figure 8).

The knowledge on roles of regulatory contexts in mRNA for translation efficiency as well as the combinations of these contexts will require improvement of both experimental approaches and theoretical algorithms. The researchers must have the opportunity not only to precisely determine the correlation between the observed fluctuations in expression of a transcript and the actual content of the corresponding protein in plants, but also to precisely define and estimate the contributions of individual regulatory contexts and their combinations within mRNAs. Correspondingly, the need for development of an integrated information resource for this purpose looks very reasonable. This resource would comprise the information about (i) the experimental methods for assessing the changes in translation on a genome-wide scale, including their modifications and applicability to different plant species; (ii) the resources for analyzing, interpreting, and visualizing the polysome and ribosome profiling data; (iii) the resources for constructing the target sequences of plant transcripts and predicting their characteristics; (iv) the methods for experimental verification of the regulatory codes of the plant transcripts involved in translation modulation; and so on. This will form the background for coordination of the numerous studies and the insight into the fine mechanisms underlying the control of biological processes at the point of translation in plants. Also it will expand the capabilities for future studies and the potential of applications of the mRNA regulatory contexts, including their use in engineering novel plant genotypes carrying the best combinations of the corresponding alleles and the generation of new of transgenes, including the use of genome editing technologies.

## Figures and Tables

**Figure 1 ijms-20-00033-f001:**
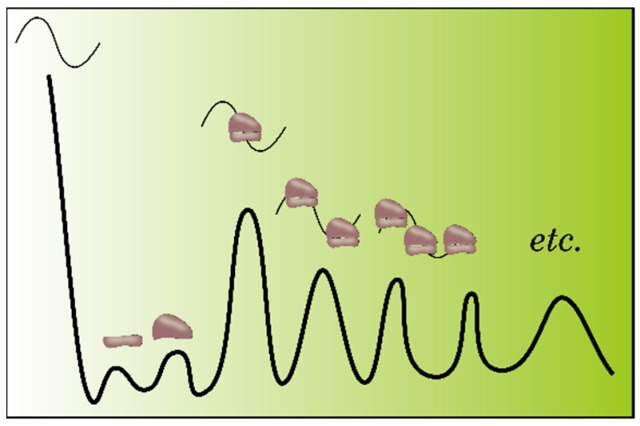
Polysome profiling in a sucrose density gradient. Separation of the transcripts depending on the ribosome loading: the first peak corresponds to the mRNAs unbound to ribosomes; second and third peaks, to the ribosome small and large subunits, respectively; and the fourth and subsequent peaks, to the mRNAs with different ribosome loadings.

**Figure 2 ijms-20-00033-f002:**
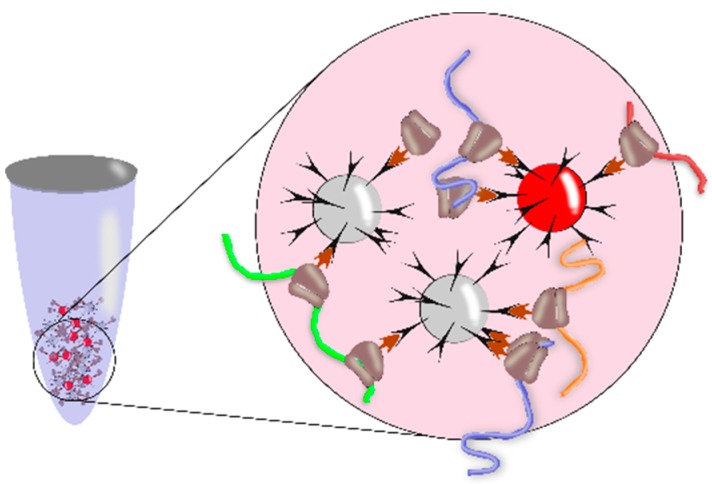
Polysome profiling using translating ribosome affinity purification (TRAP). General principle of selective separation on anti-FLAG-M2 agarose of the transcripts bound to ribosomes carrying the epitope-tagged variant of ribosomal protein. Brown arrows denote the FLAG epitope in ribosomal protein and black icons denote the anti-FLAG on agarose beads.

**Figure 3 ijms-20-00033-f003:**
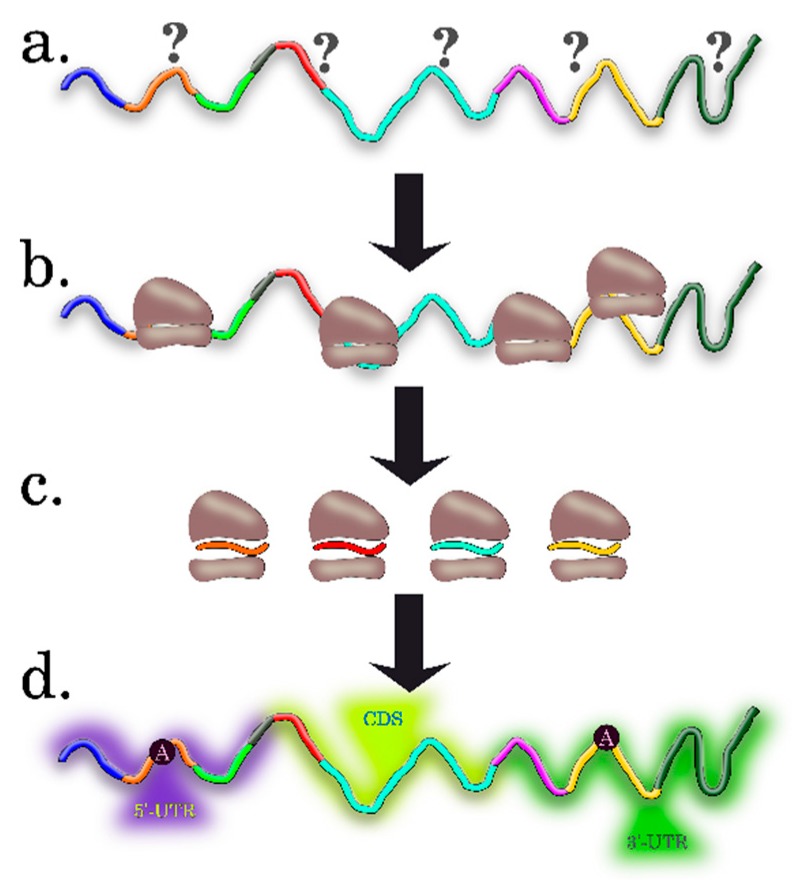
Scheme of application of ribosome profiling to functional characterization of mRNA regions. (**a**) Scheme of an mRNA with unknown ribosome positions. (**b**) The mRNA with arrested ribosomes in the transcript regions potentially important for efficient translation. (**c**) Formation of the ribosome footprints by RNase hydrolysis. The resulting footprints characterize the translational functionality of a certain mRNA region. The footprints shown with different colors correspond to different mRNA regions. (**d**) Result of analysis of the precise positions of translating ribosomes along mRNA, where A is the identified alternative open reading frames in 5’UTRs or 3’UTRs and CDS (coding sequence) is the main reading frame of the transcript.

**Figure 4 ijms-20-00033-f004:**
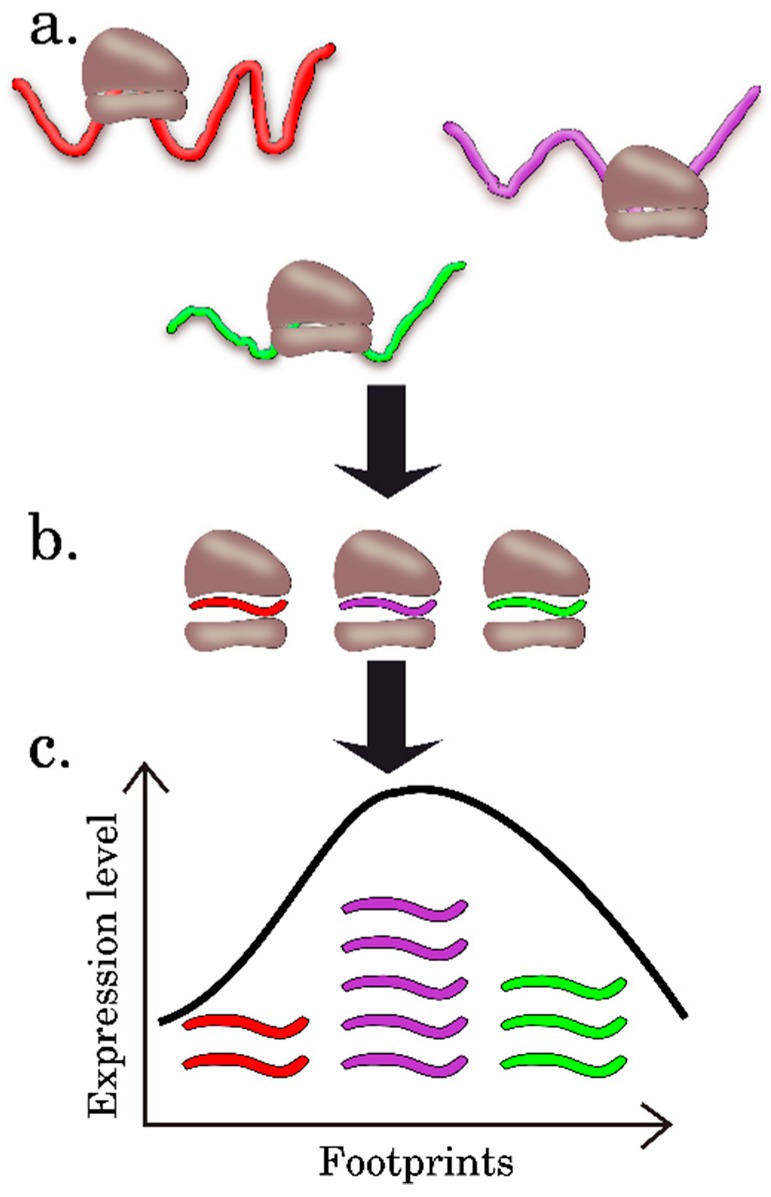
Principle of analysis, interpretation, and visualization of the ribosome profiling data. (**a**) The ribosomes arrested on transcripts (**b**) form ribosome footprints after RNase hydrolysis. (**c**) The footprints mapped onto genome can be associated with particular sequences to assess the relative amount and positions of ribosomes on the transcripts on a genome-wide scale.

**Figure 5 ijms-20-00033-f005:**
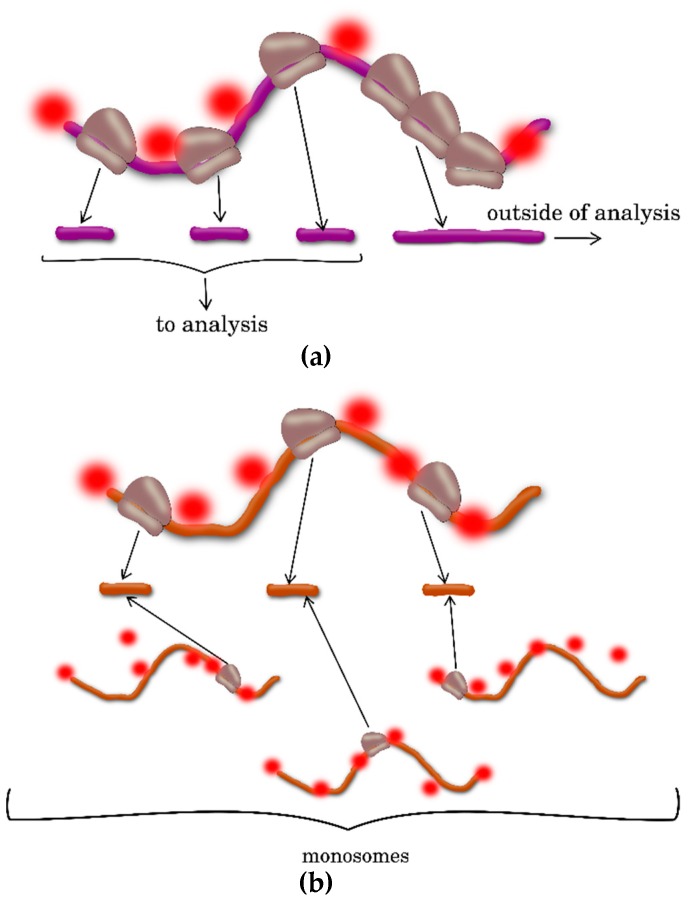
Limitations in the use of ribosome profiling: (**a**) overestimation of translation efficiency because of the footprints of monosomes, where mRNA is also protected by ribosome and (**b**) underrepresentation of the transcript region with stacked ribosomes; carefully stacked polysomes are inaccessible to RNases, thus cannot be digested into ribosome footprints of the tested size (28–30 nucleotides). Red spots denote the region attacked by RNase.

**Figure 6 ijms-20-00033-f006:**
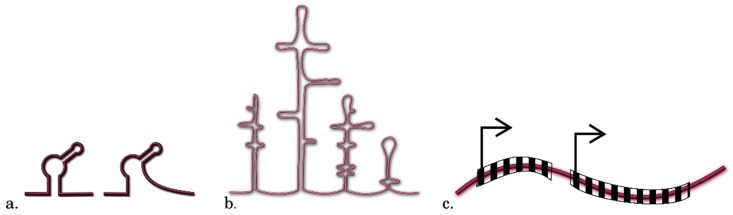
Examples of some mRNA cis-regulatory elements: (**a**) riboswitches; (**b**) internal ribosome entry sites (IRESs); and (**c**) alternative open reading frames.

**Figure 7 ijms-20-00033-f007:**
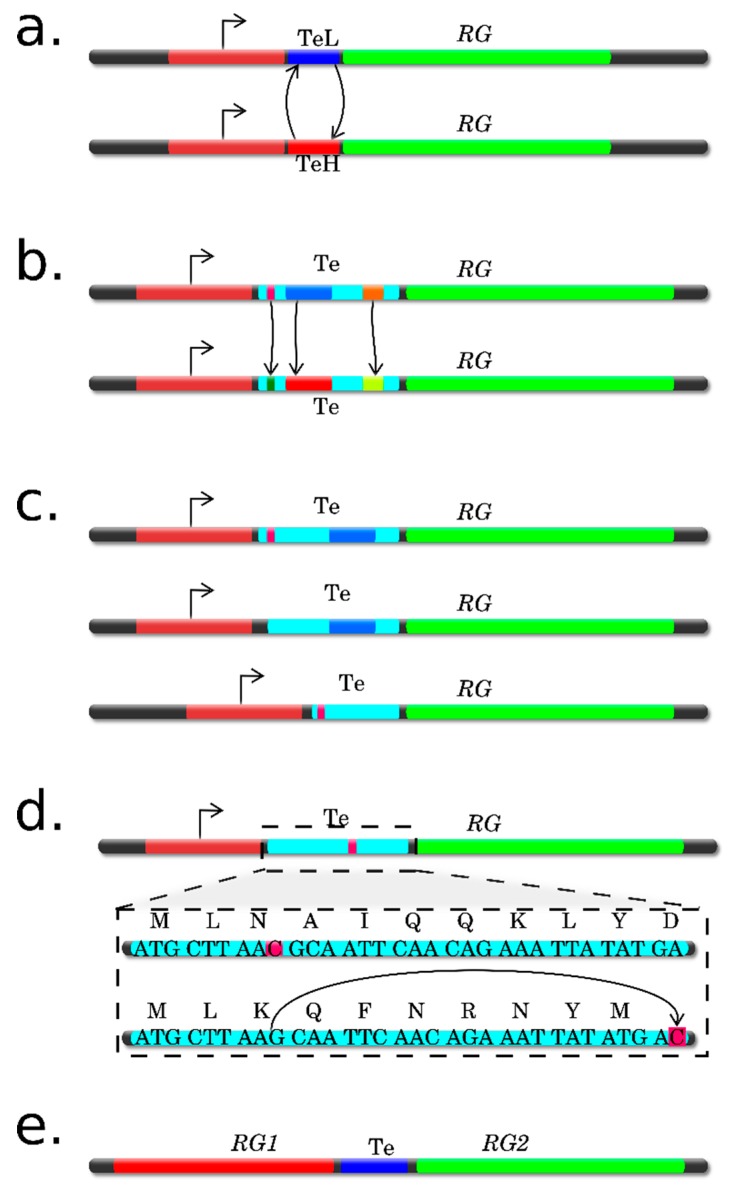
Approaches to experimental verification of the tested regulatory elements using the strategy of reporter systems. (**a**) Combinatorial approach. Mutual substitutions of the predicted regulatory motifs TeL and TeH (denoted with arrows) are introduced into the pairs of 5’UTRs of the same length but considerably differing in the experimentally confirmed translation efficiencies. TeL and TeH are the tested elements characteristic of the transcripts with low and high translation efficiencies, respectively. (**b**) Site-specific mutagenesis. The native regulatory sequence is above and the mutant regulatory sequence is below; different colors denote the region used for mutagenesis; direct arrows indicate the substituted regions in two sequences. (**c**) Deletion analysis. The native regulatory sequence is above; different colors denote the regions used for deletions: the regulatory region with deletion in the 5’ region is in the middle (deletion of the pink region of the native sequence) and the regulatory region with deletion in the 3’ region is below (deletion of the blue region of the native sequence). (**d**) Frameshift analysis. The region for introducing frameshift is dashed; the native nucleotide and amino acid sequences are above and the mutant, below. The nucleotides colored red were frameshifted. Simultaneous introduction of deletions and insertions to positions −1 and +1 changes the amino acid composition of peptide sequence encoded by the alternative open reading frame preserving the presence of the overlapping peptide and its length. (**e**) Bicistronic construct for studying the functionality of IRESs. Two reporter genes are translated from the bicistronic construct in a coordinated manner; translation of one of them (RG2) is controlled by the tested element (Te) and the other (RG1) is translated according to the classical cap-dependent mechanism. In all panels, bent arrows denote the transcription start point; Te, tested element; and RG, reporter gene.

**Figure 8 ijms-20-00033-f008:**
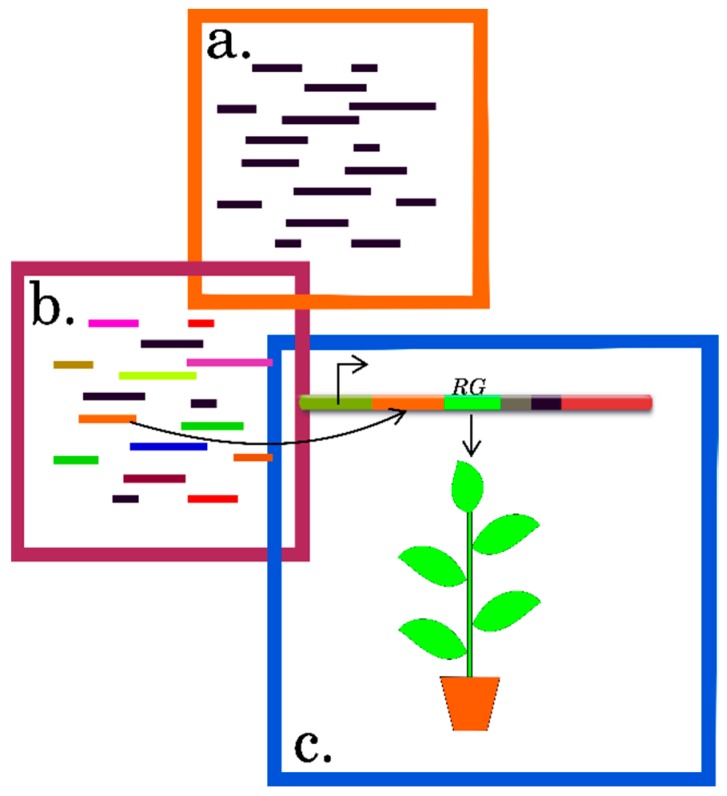
General strategy for identification, prediction, and experimental verification of the functional elements within transcripts that mediate their efficient translation. (**a**) Primary data on the transcripts differing in their translation efficiencies are obtained by sequencing the pools of the transcripts generated by polysome profiling, TRAP, or ribosome profiling. (**b**) In silico analysis of the transcript sequences identified the functional elements of transcripts. (**c**) Experimental verification of the predicted functional elements within transcripts (case study of reporter systems).

**Table 1 ijms-20-00033-t001:** Comparative characterization of the experimental approaches producing pools of differentially-translated mRNAs.

Experimental Approach	Basic Protocol	Advantages	Limitations	References
**Polysome Profiling**	Separation of transcripts with different ribosome loading by ultracentrifugation; supplemented by sequencing of different mRNA fractions, including transcriptome-wide analysis	Simplicity and possibility to analyze the plant species with annotated and unannotated genomes	Does not assess the number and location of ribosomes on each transcript	[4,5,21]
**Translating Ribosome Affinity Purification**	Separation of the transcripts with different ribosome loadings by absorption on anti-FLAG-M2 agarose; supplemented by mRNA sequencing, including transcriptome-wide analysis	Profiling of actively-translated RNAs from different plant tissues and particular cell types; Identification of only the mRNAs bound to ribosomes, which makes it possible to avoid the potential confusion with the transcripts associated with the other RNA-binding proteins	Cannot estimate the number and location of ribosomes on each transcript; Requires production and accurate selection of the plant transgenic lines that express an epitope-tagged variant of ribosomal protein L18	[3,12,14,15]
**Ribosome Profiling**	Isolation and sequencing of the ribosome-protected mRNA fragments; modification of the protocol is necessary for individual species	Identification of the ribosome number and location on each transcript; Detection of new translated ORFs and noncanonical translation start sites	Requires significant material, time and labor investments; A considerable amount of biological material is necessary; The transcript regions with stacked ribosomes may be underrepresented; Incorrect trace identification may result from the RNA interaction with RNA-binding proteins of an analogous size; Applicable only to the plants with a well-annotated genome	[4,18,19,20,24]

Note: The key advantages and limitations are shown for each approach; see the text for a comprehensive description.

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
