# Peer review of "Computational and Experimental Tools to Monitor the Changes in Translation Efficiency of Plant mRNA on a Genome-Wide Scale: Advantages, Limitations, and Solutions"

_ijms, 2018, doi:10.3390/ijms20010033_

Reviewer 1 Report

The manuscript is very interesting and useful for researchers, large amount of information has been analyzed.

Some comments and suggestions:

- The language should be improved, in some places 'non-scientific' terms are used (e.g. 'fate' in line 56, 'dwells' in line 66 etc.). I would suggest to avoid such terms in scientific publication. Some language constructions are not very successful (e.g. ', such as, for example' in line 38 etc.). Therefore authors should improve the language of manuscript.

- The chapter '2. Experimental Approaches...' is logically split in sub-chapters for better understanding. In turn, chapter '3. Computational Algorithms... ' is presented as one block - it is quite complicated to read and follow to the discussion. I would suggest to split also this chapter in logical parts.

Author Response

Thank you very much for reviewing our manuscript. We are grateful for the thoughtful comments from the reviewer. We did our best to resolve all the problems that you indicated in the revised manuscript version. English was significantly improved, including 'non-scientific' terms, noted by the reviewer, as well as unsuccessful language constructions. We also introduced subsections into chapters 3 and also 4. We hope that all concerns were adequately addressed and we believe that the revision allowed for a considerable improvement of the manuscript.

Yours sincerely,

Irina Goldenkova-Pavlova

Reviewer 2 Report

SUMMARY

The manuscript by Goldenkova-Pavlova et al. is review of the experimental methods for genome-wide analysis of translational control, computational algorithms applicable for searching and analyzing the various regulatory contexts within mRNAs, and the approaches for verifying roles of the regulators in mRNA translation in plants. Regarding the topic, the review is clear at where we are at currently, and where the future direction is pointing. Moreover, most of the important studies in this area of research are included in the literature. Having said this – I have minor general and specific comments for the authors’ consideration.  

GENERAL COMMENTS/SUGGESTIONS

There are too many important information under individual “Headings/Titles”; and this can make readers easily lose track of the various information being discussed. Therefore, I suggest the authors try to include “Subheadings/Subtitles” wherever necessary. Secondly, there are too many paragraphs that I believe can be merged. I made some few suggestions on this in the “Specific Comments/Suggestions” below. Moreover, some of the paragraphs are just to short to stand by themselves as paragraphs.

SPECIFIC COMMENTS/SUGGESTIONS

Line 14: You may want to consider replacing the word “decisive” with more appropriate words like “vital” or “necessary” or “required”. Ditto - you may also want to consider replacing the word “decisive” in Line 706. Furthermore, in the same sentence in Line 706 – do you mean “….RNA capable of switching between …….” instead of “……RNA able to switch between…..”?

Line 18-21: Please cross-check the sentence starting from Line 18 and ending in Line 21. It lacks clarity.

Line 90: “utile” or “futile” or “utilize”? Ditto in line 410. Please confirm.

Line 125: “The polysome profiling appeared rather efficient in the studies on differential translation …..” instead of “The polysome profiling appeared rather efficient in the studies into differential translation ….” i.e. replace [into] with [on]

Line 144: Replace “depression” with “repression”

Line 185: Italicize “A. thaliana”. Ditto in Line 209. 298, 308, 314, 339,

Line 213: Italicize “in vivo”. Ditto in Line 216

Line 227 – 228: Italicize “Medicago truncatula”. Same goes for “N. benthamiana” in same sentence

Line 361-362: “….this is associated with that the meticulously stacked polysomes may prevent hydrolysis into monosomes because of inaccessibility to RNases (Figure 5B)” Please cross check the sentence to see if a word is missing between […..associated with] and [that the meticulously stacked……]

Line 413: You may want to consider “In this section, we summarized the resources ……” instead of “In this section, we brief the resources….”. And if (in Line 413) the authors are referring to only “some of the resources and algorithms” then this needs to be included in the suggested sentence. Ditto in Line 837: Consider replacing “briefed” with “summarized”

Combine paragraphs 507-516 and 517-522 into one paragraph; Ditto for paragraphs 523-526 and 527-529.

Line 596: Replace “expectable” with “expected”

Line 624: Replace “consider” with “considered”?

Line 654: Do you mean “47 genes”? In other words – should you use the plural for gene (i.e. with an “s” at the end”)?

Line 656: “….when constructing plant producers of biologically active…..” Do you mean “…..when engineering plants producing biologically active….”?

Line 707-711: “Note that an important specific feature of the riboswitches is their ability to both activate and inhibit translation from the controlled ORF thanks to the presence of a specific regulatory region, aptamer, which is part of the transcript, frequently with a developed secondary and sometimes even tertiary, structure conferring on it the properties of a receptor for small molecules (ligands).” I find this sentence confusing. Splitting it into two separate sentences will be helpful to readers.

Line 748: “retains” instead of “retain”

Line 754: “Moreover, it has been convincingly……..” instead of “Moreover, is had been convincingly……” i.e. replace [is] with[it]

Line 746-774: Combine the two paragraphs.

Line 794: Consider replacing “united” with “ligated”

Line 875: “…….the need for development of an integrated…….” instead of “…….the need in creation of an integrated…….”

Author Response

Thank you very much for the careful review. We are especially grateful for the detailed comments of the reviewer. We agree to all the suggestions and improved the manuscript accordingly. We introduced subsections into chapters 3 and also 4. English was also significantly improved. We believe that in the current state the manuscript fully addresses all comments of reviewer.

We hope that all concerns were adequately addressed and we believe that the revision allowed for a considerable improvement of the manuscript.

Yours sincerely,

Irina Goldenkova-Pavlova